# Tensions and Paradoxes of Scaling Up: A Critical Reflection on Physical Activity Promotion

**DOI:** 10.3390/ijerph192114284

**Published:** 2022-11-01

**Authors:** Harriet Koorts, Adrian Bauman, Nancy Edwards, William Bellew, Wendy J. Brown, Mitch J. Duncan, David R. Lubans, Andrew J. Milat, Philip J. Morgan, Nicole Nathan, Andrew Searles, Karen Lee, Ronald C. Plotnikoff

**Affiliations:** 1Institute for Physical Activity and Nutrition (IPAN), School of Exercise and Nutrition Sciences, Deakin University, Geelong, VIC 3220, Australia; 2Prevention Research Collaboration, Sydney School of Public Health, The University of Sydney, Sydney, NSW 2006, Australia; 3The Australian Prevention Partnership Centre, Sax Institute, Sydney, NSW 2037, Australia; 4School of Nursing, University of Ottawa, Ottawa, ON K1H 8M5, Canada; 5Sydney Medical School & Sydney School of Public Health, The University of Sydney, Sydney, NSW 2006, Australia; 6School of Human Movement and Nutrition Sciences, University of Queensland, St Lucia, QLD 4072, Australia; 7School of Public Health, University of Newcastle, Callaghan, NSW 2308, Australia; 8Centre for Active Living and Learning, College of Human and Social Futures, University of Newcastle, Callaghan, NSW 2308, Australia; 9Hunter Medical Research Institute, New Lambton Heights, NSW 2305, Australia; 10Faculty of Sport and Health Sciences, University of Jyväskylä, 40014 Jyvaskyla, Finland; 11Centre for Epidemiology and Evidence, NSW Ministry of Health, 1 Reserve Rd., St Leonards, NSW 2065, Australia; 12School of Medicine and Public Health, The University of Newcastle, Newcastle, NSW 2308, Australia; 13Hunter New England Population Health, Hunter New England Area Health Service, Newcastle, NSW 2287, Australia

**Keywords:** physical activity promotion, scale-up, implementation science, interventions, research scalability, scalable interventions, scale-up barriers, scale-up facilitators, health impact, population health

## Abstract

Background: Achieving system-level, sustainable ‘scale-up’ of interventions is the epitome of successful translation of evidence-based approaches in population health. In physical activity promotion, few evidence-based interventions reach implementation at scale or become embedded within systems for sustainable health impact. This is despite the vast published literature describing efficacy studies of small-scale physical activity interventions. Research into physical activity scale-up (through case-study analysis; evaluations of scale-up processes in implementation trials; and mapping the processes, strategies, and principles for scale-up) has identified barriers and facilitators to intervention expansion. Many interventions are implemented at scale by governments but have not been evaluated or have unpublished evaluation information. Further, few public health interventions have evaluations that reveal the costs and benefits of scaled-up implementation. This lack of economic information introduces an additional element of risk for decision makers when deciding which physical activity interventions should be supported with scarce funding resources. Decision-makers face many other challenges when scaling interventions which do not relate to formal research trials of scale-up; Methods: To explore these issues, a multidisciplinary two-day workshop involving experts in physical activity scale-up was convened by the University of Newcastle, Australia, and the University of Ottawa, Canada (February 2019); Results: In this paper we discuss some of the scale-up tensions (challenges and conflicts) and paradoxes (things that are contrary to expectations) that emerged from this workshop in the context of the current literature and our own experiences in this field. We frame scale-up tensions according to epistemology, methodology, time, and partnerships; and paradoxes as ‘reach without scale’, ‘planned serendipity’ and ‘simple complexity’. We reflect on the implications of these scale-up tensions and paradoxes, providing considerations for future scale-up research and practice moving forward; Conclusions: In this paper, we delve deeper into stakeholders’ assumptions, processes and expectations of scaling up, and challenge in what ways as stakeholders, we all contribute to desired or undesired outcomes. Through a lens of ‘tensions’ and ‘paradoxes’, we make an original contribution to the scale-up literature that might influence current perspectives of scaling-up, provide future approaches for physical activity promotion, and contribute to understanding of dynamic of research-practice partnerships.

## 1. Introduction

Population health improvement requires that interventions are delivered equitably, effectively and sustainably at scale [1,2]. Efficacy trials are often conducted to test the effects of health interventions, which are trials conducted under ‘optimum conditions of program implementation and recipient participation’ [3]. For population-level interventions implemented in ‘real-world’ conditions at scale, however, effectiveness trials are often more suited, which are trials conducted under less controlled ‘real-world’ conditions, where variable implementation and uptake can be expected [3]. Scale-up is defined as ‘replicating and extending the reach of an intervention into other localities, cities, or regions’ [4], involving ‘deliberate efforts to increase the impact of innovations successfully tested in pilot or experimental projects so as to benefit more people and to foster policy and program development on a lasting basis’ [5]. Scale-up can be considered ‘successful’ once an intervention (programs, strategies, policies or initiatives) achieves system-level embeddedness and sustainable health impact [6].

Scaling up evidence-based interventions has been ubiquitously difficult in public health. Systematic reviews have identified barriers to scaling-up public health programs including: a lack of policy/political support, lack of funding and other resources (e.g., human and infrastructure) for scaling-up and a perceived lack of need for the proposed intervention [7]. Whilst there are some successful scale-up examples in public health (e.g., reproductive health [8] and HIV testing [9]), for complex non-communicable disease risk factors, such as physical activity [10], few research-led interventions have achieved sustainable delivery and impact at scale [6]. Globally, this remains problematic; despite significant investment by governments internationally, physical inactivity levels remain high [11], contributing to healthcare costs in excess of INT$50 billion [12]. The scale-up of effective physical activity interventions is essential if we are to achieve population-wide health improvements. For example, in Brazil, Project GUIA (a cross-national academic-government partnership) led to the successful scaling up of the effective ‘Academia da Saúde’ (involving physical activity classes in community settings) to 4000 cities in Brazil [13]. Amongst other things, a national and international network of researchers, rigorous methodology and political support and partnership were essential factors for its ongoing national implementation at scale.

Over the last few decades the World Health Organization (WHO) and other international consortia have urged policy makers and researchers to focus their efforts on scaling-up evidence-based physical activity programs [4,14]. For example, the WHO Global Action Plan on Physical Activity (GAPPA) emphasises the need for country-level system reforms to facilitate scalable solutions to physical inactivity [15]. Despite promising progress, translating and sustaining effective interventions into large-scale settings remain challenging [16,17,18,19]. In the scientific literature, scale-up barriers include fiscal difficulties, lack of political investment [7] ethical implications associated with exacerbated inequities, a lack of involvement from target users, and strategic connections between stakeholders and policy priorities [20]. In physical activity, interventions are often designed without sufficient consideration for real-world implementation and scale-up [21], and lacking theoretical grounding in implementation frameworks that can facilitate real-world translation [22]. The task of accelerating implementation of scalable solutions to physical inactivity is needed, although this requires reliable, dedicated resources (both human and fiscal), and establishment of strategic connections between stakeholders and policy priorities [15]. Nonetheless, gaps remain in the literature in understanding why some interventions are successfully scaled over others, and how we can facilitate the scale-up process.

To explore these issues and individuals’ experiences of scaling up physical activity interventions, in 2019 an invitational two-day scale-up workshop, co-hosted by the University of Newcastle, Australia, and the University of Ottawa, Canada, was convened for academics, practitioners and policymakers (stakeholders) involved in scaling up physical activity interventions. Drawing on the issues which emerged during this workshop and reflecting on them in light of the current literature, this paper discusses a number of tensions (challenges and conflicts) and paradoxes (things that are contrary to expectations) stakeholders face when scaling up physical activity interventions. We attempt to push the scale-up field forward by delving deeper into stakeholders’ assumptions, processes and expectations of scaling up, and challenge how we all contribute to desired or undesired scale-up outcomes. We discuss the implications of these issues for future scale-up research and practice for the physical activity field.

## 2. Materials and Methods

### Physical Activity Scale-Up Workshop

An invitational two-day workshop was held at the University of Newcastle, Australia in February 2019, for a group of multidisciplinary scale-up experts (*n* = 27) from Australian and Canadian academic institutions and government/non-government organisations. The 27 experts were identified via the existing networks of NE and RP related to physical activity scale-up, and invited by email to take part. Objectives of the workshop were to: (1) identify funding, evaluation and other challenges to scale-up of physical activity interventions during scale-up; (2) consider research designs, optimal process and outcome measures, and costing approaches during scale-up; and (3) enhance research collaboration between government and academic organisations for future scale-up of physical activity interventions. Day one of the workshop focused on challenges experienced when scaling and what success looked like. Workshop facilitators captured group discussions, independently, via session notes, which were summarised and grouped into core topic areas (by NE and RP). Day two of the workshop involved case study presentations and small group discussions to illustrate different research designs suited to implementation and scale-up research, and ways of conducting economic analyses and costing future implementation. After completion of the workshop, during the manuscript write up phase, the core topic areas identified by NE and RP were organized and framed by HK, as ‘tensions’ and ‘paradoxes’ in the scale-up field. These tensions and paradoxes provide the structure for this paper, which we discuss in the light of the current literature.

## 3. Results

### 3.1. Tensions in Scale-Up

Tensions were identified as challenges in scaling-up physical activity interventions. Tensions were seen as an inevitable consequence of the diverse mix of individuals and organisations needed for research translation, and the competing expectations of researchers and practitioners/policymakers. We frame the key tensions discussed in the workshop according to four domains: (i) epistemology (sources of knowledge and ways of learning about social reality); (ii) methodology (ways of carrying out research); (iii) time (required for scaling up); and (iv) partnerships (expectations and ownership).

### 3.2. Epistemological Tensions

Epistemology relates to the nature of knowledge and different ways of gaining knowledge, which contrasts with ontology that relates to reality and what exists or does not exist. Epistemological tensions were discussed were related to two concepts. First, ‘ways of learning about social reality’ describes the contradictions between scientific efficacy paradigms of evidence generation to inform scale-up actions, and the differing influences on individuals’ conceptualisations of successful scale-up. Second, ‘sources of knowledge’ related to the contrast between the scientific writing about the translation pipeline and stakeholders’ experiences in practice.

#### 3.2.1. Paradigms of Evidence Generation and Conceptualisations of ‘Successful’ Scale-Up

Interdisciplinary stakeholders bring unique perspectives to the scale-up process based on knowledge derived from experience as to how scale-up should (and can or has) occur(red). Efficacy studies often fail to consider the subsequent scale-up of interventions, or their potential need for adaptation in practice [21,23,24,25,26,27]. This results in abundant evidence-generation but many translation failures. A scientific paradigm might suggest that adaptation, fidelity and other elements of implementation should be tested in controlled trials, but this level of research control may not be feasible in real-world scale-up of physical activity interventions. 

A ‘scale-up paradigm’ was described in the workshop as interventions being more likely to be re-shaped and modified by users, perhaps reducing intervention ‘voltage drop’ [28]. In reality, scale-up paradigms are closer to ecological frameworks and systems perspectives [29]. In this systems approach, implementation is considered an iterative, dynamic, adaptive and non-linear process that does not easily conform to narrow perspectives on fidelity [17,30,31]. Yet, in physical activity, whilst a systems thinking approach was considered somewhat important, it was also perceived as less feasible to achieve [6]. The authors suggest that this gap between perceived ‘importance’ and ‘feasibility’ in scale-up, may impede effective action. If the field is to adopt new paradigms and perspectives on scale-up, new ways of thinking about physical activity programs and their sustainment are required. 

#### 3.2.2. Literature Depictions of Translation versus Scale-Up in Practice

Our workshop discussions identified a tension between the processes of research translation and the ways in which researchers tackle scaling up. Several theoretical frameworks describe research translation as a continuum from efficacy to scale-up (e.g., [32]), including how implementation research can feature along this continuum (e.g., [33]). Much of the information on scale-up has been through case-study evaluations (e.g., [34]), studies of intervention voltage drop as studies are scaled up (e.g., [35]), and studies of factors influencing scale-up processes and strategies (e.g., [7]).

This literature has contributed to advancing frameworks relevant to scale-up (e.g., the NASS framework [36]), informing scale-up guides for practitioners and policymakers [37], planning tools for academics and stakeholders [21] and instruments for assessing intervention scalability [38]. Multiple perspectives have been used to study scale-up processes, with implementation science (study of methods to promote uptake of evidence in practice) being the traditional approach within the scale-up literature [29]. As with the pipeline model of research translation, implementation science tends to describe the linear unfolding of interventions with an emphasis on fidelity through a planned approach [29]. Scaling-up in physical activity has predominantly adopted this sequential ‘intervention-oriented’ approach [16], which contrasts with complex systems perspectives to understand implementation [39,40,41] that may be more appropriate when scaling up [16,42]. Workshop attendees described real-world examples of the design, testing, adaptation and wider implementation of interventions that were non-linear. For example, a study of scale-up pathways among 40 public health interventions showed that research translation varied, and that not all stages of the linear pathways were necessary for an intervention to be scaled. [43]. At times no previous, published, evidence of effectiveness was evident in prevention projects that were scaled-up.

### 3.3. Methodological Tensions

Our discussions also identified methodological tensions related to the diverse perspectives of studying scale-up and use of appropriate study designs. Managing contextual adaptions during scale-up and challenges capturing appropriate outcomes at scale (intended and unintended) was also a source of tension.

#### 3.3.1. Use of Appropriate Research Designs for Scale-Up

Many researcher-led interventions are too complex for integration into routine delivery settings, and hence fail to be effectively translated [23]. Measures for studies of effectiveness may not capture the breadth of factors influencing scale-up. Economic evaluations, when they are undertaken, often use Quality Adjusted Life Years (QALYs) as the established measure of benefit. These analyses reveal the efficiency of the intervention but fail to reflect equity considerations [44]. In efficacy and observational studies, for example, there are standardised reporting systems and criteria to benchmark the quality of study design, such as the CONSORT statement [45] and the STROBE statement [46], respectively. In implementation research, the Standards for Reporting Implementation Studies (StaRI) Statement [47] guides the reporting of implementation studies and trial designs specifically tailored for capturing implementation (e.g., hybrid effectiveness-implementation trials [48]). Whilst there are real-world pragmatic trials [33] and participatory research approaches [49], and frameworks such as the behaviour change wheel [50] and RE-AIM [51] that are often used to design and evaluate outcomes of interventions and implementation, respectively; scale-up does not have a single recommended design or evaluation approach. Recent suggestions include systems thinking approaches and engaging decision-makers to increase the likelihood of practice impact [52].

#### 3.3.2. Contextual Adaptation during Scale-Up

There was consensus during the workshop regarding the importance, and inevitability, of contextual adaptation of physical activity interventions/implementation when scaling up. Adaptation was a source of tension for some academics, as they had to “let-go” of the intervention as it took hold in the system. A lack of transferability of interventions may potentially exacerbate inequalities in physical activity participation [53]. There have been calls for implementation research to adopt an explicit focus on equity when studying research-practice translation [54]. For example, in Australia, government schemes, such as the ‘NSW Active Kids voucher’ (a state-wide voucher program targeting school-aged children to reduce registration costs of structured physical activity programs) have been identified as at risk of widening inequities in physical activity [55]. Whilst the scheme may alleviate the costs of participation for some families, compared to families with higher socio-economic status groups, families living in low socio-economic areas were less likely to adopt the scheme due to lack of ongoing funding support post intervention. 

Moving from efficacy to effectiveness/replication to scale-up studies may require alternative delivery strategies or systems, to new communities or target groups. Tensions in contextual adaptation may reflect the traditional research view that fidelity is obtainable and essential during real-world implementation. The dynamic sustainability framework, however, describes intervention adaptation as a continuous improvement process that is linked to sustainability [28]. Nonetheless, the effects of adaptation on intervention effects can be mixed. For example, in school-based physical activity research, level of adaptation (fidelity) may not be associated with program efficacy [56] or lead to reduced effects [57]. This highlights the importance of enabling and integrating ways for intervention components to be adapted and monitoring the associated outcomes. 

It is not always clear whether transfer failure is due to poor generalisability of the efficacious intervention, or to other problems with the methodological evaluation [58]. There are tools to classify adaptations of evidence-based interventions [59]. Planning for adaptations requires working with stakeholders to identify which intervention/implementation components or principles are essential and be adapted during scale-up (‘core components’) and which are flexible to enable implementation in diverse settings [21]. 

#### 3.3.3. Capturing Appropriate Outcomes at Scale

Workshop attendees explained that the number of settings, contexts and systems affected during scale-up can extend beyond the capacity for data collection. Long-term sustainability considerations included whether relevant administrative databases or other existing information systems could be used to provide ongoing and long-term monitoring and feedback information to inform intervention scale-up (e.g., [60]). The workshop considered to what extent monitoring of outcomes (intended and unintended) is possible at scale and how this contrasts with the level of precision necessary for stakeholders. 

At scale, information systems can enable a shift from static, retrospective measures of implementation (aligned more with process evaluation) to ongoing monitoring and benchmarking potential. Research designs that can capitalize on pre-existing data collection processes within settings (i.e., population monitoring data that may be collected as part of local government reporting requirements) may assist with capturing reach and effectiveness at scale. However, capturing both impact and reach can be challenging. In Australia, State and Federal governments have supported implementation and evaluation of scaled-up obesity prevention initiatives. For example, in the Victorian state program, ‘Healthy Together Victoria’ (HTV) a whole-of-systems approach to prevent childhood obesity (2011–2016); the South Australian government led ‘Obesity Prevention and Lifestyle’ (OPAL) initiative (2009–2017); and the NSW government ‘Healthy Children Initiative’ (HCI) 2011–2020; evaluation was embedded in the scale-up planning process and involved assessments of implementation and impact. All three programs demonstrated high reach, yet they all experienced issues capturing impact data [61,62,63]. Challenges of capturing impact data at-scale may have contributed to the overemphasis on program reach in evaluations of scale-up.

Whilst monitoring processes can be initiated once scale-up has commenced, there are challenges when assessing the potential scalability of interventions. This includes establishing the optimal timing, consensus on scale-up purpose, and tensions between preliminary testing and designing an intervention ‘fit-for-scale’ [64]. Conflict between program funders and researchers may be due to differences in the perceived importance and value of evaluation data and different outcomes [65]. For physical activity and nutrition interventions scaled in Australia, Government sets variable standards for evidence that is ‘fit-for-purpose’ to assist with government decision-making [17]. Critically, stakeholders value different types and sources of evidence [66]. This highlights the need for detailed planning discussions of the values placed on evidence, identifying what is appropriate for different audiences, and the importance of measures relevant for the sector(s) and stakeholder(s).

### 3.4. Tensions of Time

Attendees referred to the ‘scale-up time lag’ from initiatives being fully tested and their subsequent scale-up. Conversely, urgency in scale-up within limited time frames posed challenges for academics. Further, funding cycles between academic funding bodies and municipal government programs is often incongruent. The time required for scaling up was also perceived as particularly problematic for digital health interventions, which were competing with fast-paced developments in technology [67].

#### Scale-Up ‘Time Lag’ Impacting Decision-Making in a Rapidly Changing Environment

Whilst it is known that advanced planning for scale-up enhances potential outcomes [68], the time lag often means that partner organisations may need to proceed with scale-up before all the evidence generation work is completed or optimal modifications have occurred. Scaling an intervention, from pilot to national implementation, may take up to 15 years [69], and this process requires sustained investment of funds and political support [70]. Policymaker impatience may stem from changes in political priorities, whereas researcher impatience may stem from policy delays in funding and/or legislation/regulations, and the time spent waiting for the opportunistic ‘scale-up window’ [71,72]. The scale-up time lag and ‘messiness’ of real-world research was discussed as a disincentive for early career academics who are rewarded for ‘productivity’ in terms of quantifiable outputs (i.e., peer-reviewed manuscripts). An international study of barriers to conducting dissemination and implementation research in physical activity reported that for early career academics real-world research may be “career suicide” and students were discouraged from pursuing implementation science and scale-up work with stakeholders [73].

### 3.5. Partnership Tensions

Tensions within partnerships related to unclear decisions on ‘ownership’, costs and responsibilities for the scaled-up intervention and its evaluation. 

#### 3.5.1. ‘Ownership’ and Responsibility for Scale-Up

A key consideration for scale-up is who ‘owned’ the intervention being scaled once it has moved beyond the boundaries of researcher-controlled studies. Whilst the researcher generally ‘owns’ the initial ‘intervention product’, ownership shifts to the funding organisations as interventions are implemented at scale. In particular, ownership and sustainability of support became challenging if there were changes in Government, funding and political priorities. Questions about intervention branding, trademarks and intellectual property surfaced as academics described undertaking the transition to scale-up. Program identity and ownership is also required to sustain programs beyond the initial scale-up phase. Partners who perceive an ownership stake in research that reflects their priorities, are key to sustainable scale-up [74]. Sustainability was largely seen as the domain of partners, and it raised the issue of identifying system-based champion(s) who have a stake in the intervention and its continuation. Another point of consideration is if an intervention is not sustained, is it considered to be a scale-up failure? Discussions that focus on co-design and engagement of all stakeholders early in the scale-up planning process may assist here. Primarily, as early and active collaboration with key stakeholders through co-design, can facilitate more attainable strategies for implementation, and potentially improve scale-up outcomes. For example, a co-designed physical activity program for older adults involved delivery partners across multiple levels of implementation, to co-create a tailored approach to enhance implementation at scale [75].

#### 3.5.2. Funding, Costs and Benefits of Scale-Up

Economic arguments can influence government decision-making and yet there is limited evidence of using economic analyses to inform the best-buys of scaled up population-based physical activity programs [44,76]. Attendees described various ways to advocate for scale-up, including aligning resource with policy priorities, providing information on program costing, and developing ongoing performance indicators. Further, narratives and success stories were considered useful for making the case for funding to decision-makers. However, for some interventions, high-level support is required for state or national scale-up, which is in addition to local and setting level buy-in for ‘on the ground’ delivery. This means the cost–benefit ratio can shift prior to and during scale-up. Workshop attendees discussed the importance of developing a business case, including a budget impact analysis [44], that builds a clear value proposition with equitable reach in mind.

Economies of scale anticipated from scaling-up an evidence-based health intervention are usually a drawcard for decision-makers regarding widespread roll-out [20]. However, there are no universal estimates of cost-effectiveness or cost benefit of scaled up interventions [20]. Some have considered costing at the per-person level for delivery at scale, rather than costing at the level of the whole state or region [77]. In health service research, the need for (but lack of) robust health-economic appraisal of new interventions has been labelled the ‘second gap in translation’ [78]. This gap has been identified as a reason for a lack of investment in prevention programs in countries such as Australia [79]. State-level overweight and obesity prevention programs that focus on both nutrition and physical activity in Australia can cost as much as AUD $20M (e.g., the Obesity Prevention and Lifestyle [OPAL] project in South Australia) [80] and AUD $40M for the New South Wales Healthy Children’s initiative [81], yet, there are few data reported on the true costs or cost effectiveness of scaling physical activity interventions [82,83,84]. 

#### 3.5.3. Obtaining Buy-In from Key Decision-Makers

Vertical scale-up involved working across various layers of the system(s) and using the priorities of different decision-making partners to obtain buy-in. Horizontal scale-up on the other hand requires engaging actors from different sectors, adding complexity in terms of partnership relations. Academics in the workshop described diverse experiences in securing buy-in from key decision-makers/partners. Many projects discussed during the workshop were well-positioned with evidence-based trials securing support from individuals across Local Government/parks and recreation [85], Departments of Education [86] Sport [61] and Health services [62], where the promotion of physical activity is inherent/core business in these sectors. However, this was often not the case for schools and workplace settings.

In schools, learning outcomes are core business, and therefore buy-in from schools and educational authorities require targeting these learning outcomes in conjunction with physical activity. This was shown in the Burn 2 Learn school-based intervention, which demonstrated the physical activity impact on cognitive and academic outcomes of senior school students [63]. This strategy was used for the wider implementation and roll-out of this intervention in NSW, Australia. In workplaces, there were challenges with unions and workplace administrators as the promotion of physical activity is often limited, and variability was noted in obtaining organisational support by workplace size and type (e.g., public sector, private and not-for profit) [87].

### 3.6. Paradoxes of Scale-Up in Physical Activity Interventions

‘Paradoxes’ refer to statements that appear contradictory but with exploration can emerge as true. Workshop discussions were categorised into three paradoxes: (i) reach without scale (overreliance on uptake as a marker of impact), (ii) planned serendipity (when chance interacts with timing) and, (iii) simple complexity (framing the complexities of scale-up).

### 3.7. Reach without Scale

Reach (proportion of the population who participate) and sustained program effectiveness are required for successful scale-up [3]. However, workshop attendees discussed that the sole reliance on capturing program reach does not sufficiently inform program effectiveness at a population level, given that the definition of successful scale-up requires ‘system-level embeddedness for sustainable health improvement’ [5]. In public health, reach is often over estimated [88] and yet metrics on intervention reach have tended to dominate studies of scale-up [16]. This poses a two-pronged challenge. Physical activity program impact can be reduced at scale [89], and thus whilst reach is a necessary metric, is it is not a sufficient indicator for successful scale-up.

### 3.8. Planned Serendipity

Opportunities and funding for scale-up often occurred through unplanned or unexpected government decisions [72]. Attendees discussed the extent that these fortuitous events could sometimes be pre-empted and facilitated, leading to ‘planned serendipity’. Less is known about the political-economic conditions that make scaling up both plausible and desirable, and how these condition inform what is scaled and who benefits [90]. Workshop attendees described experiences of working across different levels of government and waiting for ‘scale-up’ windows to open when the conditions were likely to support scale-up actions.

For example, in Australia, one contributing factor to the Victorian State Governments’ decision to scale the *TransformUs* initiative was that it coincided with the launch of a state-wide physical activity target [56]. In Canada, the scaling of Action Schools! BC across British Columbia gained Ministerial support as it coincided with the Vancouver-Whistler 2010 Olympic Games [91]. Whilst both interventions were shown to be efficacious in early trials and underwent substantial planning and partnership development for scale-up over a decade [92,93], the timing of scaling with fortuitous events was a contributor to securing government support.

Workshop attendees discussed the trade-off between the considered planning of scale-up which contrasted with reactivity in responses to opportune political moments. The timing and ‘confluence of pivotal events’ for government endorsement and roll-out can relate to serendipitous factors outside of the academics’ control such as Governments fast forwarded scale-up with funding allocations before academics considered the interventions to be science-ready. Planning for scale-up is recommended to anticipate potential harmful adverse effects of interventions [20], however, there is evidence that reacting to sudden policy support may increase sustainability of interventions at scale [17]. Planning and reactivity thus go ‘hand-in-hand’ during scale-up, and one cannot be ignored at the expense of the other. Advocates for scale-up will need to continue working within these fortuitous ‘scale-up windows’ to proceed successfully [72].

### 3.9. Simple Complexity

Attendees discussed the need to frame difficult and complex processes as clear actions to obtain buy-in. The use of ‘simple complexity’ as a communication tool can be beneficial for governments but is not without associated consequences. Workshop attendees discussed the trade-off between providing transparency of the likely time, costs and investment needed for scale-up, and the desire to maximise engagement from partner organisations.

There is a desire to seek a ‘silver bullet’ for population change when scaling physical activity interventions [18], and portraying scale-up as a controllable, step-by-step process. A narrow description of the organisations and sectors involved in scaling, and subsequent working in silos can further amplify the perception of simplicity [18]. In physical activity, scale-up usually requires multi-agency participation. The public health tendency to adopt a reductionist lens on the process of scaling (i.e., breaking it down and studying individual parts in a stepwise manner), has helped academics study the process but risks underestimating unknown events and the resources required for sustainable population-wide implementation. A major challenge in physical activity is the difficulty of conveying the complexity of scaling events to those wishing to attempt population-wide implementation. Specifically, it is important to identify overlaps and gaps in thinking between the different sectors involved in scaling, to better communicate translation of evidence to policymakers and communities. Whilst there is substantial literature for framing complex societal problems in a manner suitable for end users (e.g., FrameWorks Institute [94]), the simple complexity paradox highlights the need to capture and attribute system-level events and impacts, including when interventions should be de-implemented [18]. 

## 4. Discussion

### Insights and Considerations

In this paper, we reflect on some of the tensions and paradoxes experienced by academics, practitioners and policymakers involved in scaling-up in physical activity. These factors influence the ways in which scale-up occurs in research settings, in practice and in the partnership between them. 

The new knowledge generated from the scale-up workshop, and thus presented in this manuscript, extends what has already been learnt through studies of scaling up in the field. Building on previous research in this area that has provided valuable recommendations for improving scale-up process (e.g., [95]) and components of community-based scale-up (e.g., [96]), we adopted a unique approach to exploring the challenges and opportunities for scaling up in physical activity, by synthesizing multiple viewpoints and framing these through a reflexive and critical lens of ‘tensions’ and ‘paradoxes’. The originality of this paper lies in the fact that we delve deeper into stakeholders’ assumptions, processes and expectations of scaling up, and challenge in what ways as stakeholders, we all contribute to desired outcomes. Framing this knowledge as tensions and paradoxes gave us leverage to explore challenging concepts such as whether we can improve scaling up, and critically, how the systems we work in (i.e., academic or government) contribute to the circumstances we face in population health. 

We believe that integrating multiple, diverse perspectives, from multiple stakeholder levels throughout this manuscript, is a major strength of the paper, and increases the richness of the content we discuss. Our participant group represented researchers, practitioners and policymakers from multiple sectors, working as part of wider national and international networks of scientists and knowledge users in the field (e.g., Global Alliance for Chronic Diseases). It is important to acknowledge, however, that the participants involved in the workshop were a sample of experts identified by NE and RP. Whilst this was purposeful to ensure we captured expert options and experiences from the field, the views expressed in this paper are not intended to reflect the broader population of academics, practitioners and policymakers involved in physical activity scale-up. Nonetheless, whilst we do not provide an exhaustive list of all possible tensions and paradoxes, we describe real-world issues for those wishing to transition from studies of efficacy and effectiveness to those of implementation and scale-up:Broadening of theoretical approaches: Epistemological tensions demonstrate that our knowledge can co-exist in equilibrium and in conflict. There is no clear answer as to whether adopting a ‘best practice’ sequential approach to scale-up leads to greater population impact. If the field is to adopt new perspectives on scale-up, including systems approaches, then strategies to extend and enhance theoretical approaches will be necessary.Future considerations: The use of complexity and systems theory to understand and approach population health scale-up, on the basis that “we shift analysis from individual parts of a system to the system as a whole” [97]. Additionally, employ frameworks that incorporate a systems thinking perspective on the barriers and facilitators to scaling up (e.g., the PRACTIS Guide [21]) and tools to support assessing prospective scalability (i.e., ISAT [38]).Re-thinking data sources: Each scale-up process requires a matrix of data on outcomes, reach, adoption, fidelity/adaption, costs and sustainability at scale. These metrics are generated through a mix of research and performance monitoring processes. Multiple forms of data are required for governance and to inform different stakeholders-participants, implementers, decision makers/policy makers and political leaders.Future considerations: Refer to resources that describe key data sources for robust planning and evaluation of real-world implementation (e.g., CFIR [98]). Involve stakeholders in a participatory process to determine which additional evidence sources influence intervention uptake, political support and community sustainability. Obtain qualitative data on end user perspectives that can capture, for example, perceived evidence value, persuasiveness, and trustworthiness in the community [65].Co-creation for planning and design: Whilst scaling up does not have a single recommended design or evaluation approach, approaches that actively engage stakeholders from various system levels, early and throughout the research and process is essential.Future considerations: This can be achieved, for example, via co-creation, co-design and co-production [99] of evidence and interventions for scale-up. Research designs, partnerships and funding sources that can accommodate the length of time required to thoroughly evaluate impacts of scaling up (5+ years), may contribute to the generation of evidence that is more useful for evaluating scale-up outcomes.Shared values and shared evaluation approaches to scale-up: Over-reliance and dominance of measures of adoption and reach of scaled interventions misdirects our focus on what is required for true changes in population physical activity. We need a shared understanding of the values placed on different evidence when scaling, the limitations of data collection at-scale, what is appropriate for different audiences and the importance of adopting measures that are a priority for the sector(s).Future considerations: Facilitated discussions with stakeholder groups prior to and throughout the research process, such as via translational formative evaluation [100], is a way to capture nuances in differences in expectations for evidence generation and reporting. ‘PRACTIS Workshops’ (i.e., [101]) also offer a process to systematically identify and document key metrics for different groups, challenges/opportunities with data collection and values placed on different scale-up outcomes; beyond reach and adoption.Improving research-practice roles and partnerships: Contradictory expectations and value placed on different outcomes of scaling creates tensions for who is responsible for different aspects of program expansion. The *existence* of partnerships is fundamental for program integration across multiple sectors, yet the *quality* and maintenance of the partnerships is what leads to successful scale-up. The role of partnerships needs to start at the earliest planning stages of testing an intervention, involving policy, researcher and end-users in the scalability decisions. However, it’s important not to over-rely on single partnerships (either individuals or organizations) as over time, people change positions and organizational mandates shift.Future considerations: Partnership analysis tools (e.g., [102]) are a practical resource to assist with establishing, developing or maintaining partnerships in health promotion. The NSW Health Guide to scaling up [37] also provides a process for conducting a situational and stakeholder analysis, including when to consult with stakeholders.

## 5. Conclusions

These workshop reflections identified the need to understand the movement of research from discrete programs to multi-strategic portfolios, from one-off projects to longer-term sustained programs and from piecemeal downstream efforts to comprehensive upstream approaches. Unlike many scientific undertakings, scale-up challenges transcend domains and disciplines. Tensions can impede the progress of scale-up, whilst paradoxes encourage us to reframe our scaling mindset. Stakeholders face an ongoing dilemma on their role and influence over the scale-up process. Researchers face a conflict between the process of research translation and what is required to acquire appropriate skills, navigate partnership opportunities, and progress through their careers. We believe the issues explored in this workshop might influence existing perspectives of scaling-up, provide future approaches for physical activity promotion, and aid understanding of dynamic of research-practice partnerships. We encourage others to consider the applicability of the issues we discuss in this paper across other areas of health, as a way to advance our knowledge in the field.

## Data Availability

Not applicable.

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
