# Peer review of "Tensions and Paradoxes of Scaling Up: A Critical Reflection on Physical Activity Promotion"

_ijerph, 2022, doi:10.3390/ijerph192114284_

Round 1
Reviewer 1 Report
Title: Tensions and Paradoxes of Scaling up: A Critical Reflection on 2 Physical Activity Promotion
Overall – I believe that this is an interesting piece that offers an overview of the issues related to scaling up physical activity. The authors addressed an important research question and in doing so, they contributed to adding important evidence to this field of research.
Abstract – The abstract very well summarises the article.
Pag. 1, line 40: add “(1)” before Background, as the authors did for Methods, Results, and Conclusions.
Introduction – The introductory session provides a clear understanding of the topic and identifies the gap in the literature regarding this research area. It is very well framed. A clear rationale for the study and a clear statement of purpose were presented.
Methods – Explanations of the methods and the recruitment are presented and seem appropriate for this study. However, despite the authors mentioning that the topic areas were derived from the notes taken by the workshop facilitators, I would feel that a further description of the methodology that they used to analyse the workshop notes would be beneficial. Did the authors take notes independently and then compare and summarise them? I would be inclined to believe that the authors independently took the notes during the workshop and then they used a sort of thematic analysis approach to identify the core areas. If so, it would add a great value to this methods section if the authors could provide a brief description (even one sentence) of the approach (i.e., how the notes were taken and the areas were identified).
Results – The results are clear and describe some of the challenges in scaling up physical activity interventions. The explanation of the core themes that emerged from the workshop was clearly presented.
Pag. 6 line 283: remove comma “research in physical activity reported that for early…”
Discussion – The discussions link well with the research objective and the results, and the authors provided interesting considerations for future policy and practice implementations.
Pag. 9, lines 434-346, the authors briefly presented the strengths and limitations of this study. Perhaps, further limitations and strengths could be discussed here.
Reviewer 2 Report
The paper is well written. This is an important topic for those in physical activity promotion. The authors are encouraged to consider next steps or recommendations to move forward. The authors should also look to provide a clear indication of how this paper builds upon and adds to the research on scale-up which is cited in the paper (e.g., refs 6, 17, 18, 19). It remains unclear what this paper adds that is not already known. References for two additional and recent studies on the topic of scale-up are provided at the end of the review. Consideration of these should also be made in the case for the fit of this article. Specific comments follow.
p. 10, line 477-480: Two of the authors conducted the workshop with other authors participating. Given n=27 experts participated in the workshop, how does the authorship reflect the involvement of experts and awareness by all involved in the workshop of the production of this paper?
-p. 2, line 71: Introductory paragraph grounds readers immediately in use of terms (thank you!). Consider differentiating efficacy from effectiveness studies in this section as well.
-p. 2, line 81: In the context of goal setting, Swann and Rosenbaum (2018; doi:10.1136/bjsports-2017-098186) identify physical activity as a complex behaviour. Is this a useful distinction in addition to PA being labelled a “non-communicable disease risk factor”? Physical activity is different from other health behaviours due to its complexity.
- p. 2, line 87: I’m left wondering what examples are of “effective physical activity interventions” that could be scaled-up? What do systematic reviews/meta-analyses reveal by way of intervention effect sizes? How is ‘effective’ defined?
-p. 3, line 121: Was this paper a planned output of the workshop or developed after the fact?
-p. 3, lines 118-134: How many experts were invited (total N)? How were experts identified for the workshop? Could experts have been missed?
-p. 3: To what degree are the findings presented a result of the structure of the workshop (and focus on objectives) vs. what emerged out of the workshop itself with the expert group present? Is this a deductive summary of discussions of each workshop objective?
-p. 3, lines 144: Consider defining epistemology (in contrast with ontology) for readers. Are these tensions of postpositivist vs. critical realist vs. social constructivist worldviews?
-p. 4, lines 160+: Is there a scale-up framework/model that is recommended for health behaviours, including PA? How was “ a systems thinking approach considered achieveable [5]”? What would this look like in the real-world?
p. 4, line 170-193: The end of this section does not connect to the first part with research skills, partnerships, careers etc. The focus seems on the “tackl[ing] of scale-up’ only.
p. 5, line 213: Did workshop participants develop possible recommendations for design/evaluation? Are there any parallels in scale-up to the challenges of those engaged in natural experiments? For example, see: Scott T. Leatherdale (2019) Natural experiment methodology for research: A review of how different methods can support real-world research, International Journal of Social Research Methodology, 22:1, 19-35, DOI: 10.1080/13645579.2018.1488449
What, if any role do frameworks such as RE-AIM (e.g., Glasgow RE, Estabrooks PE. Pragmatic Applications of RE-AIM for Health Care Initiatives in Community and Clinical Settings. Prev Chronic Dis 2018; 15:170271. DOI: https://doi.org/10.5888/pcd15.170271) or the Behaviour Change Wheel (Michie et al., 2014) play in scale up or ‘real world’ interventions or in addressing some of the tensions described?
p. 5, line 221: Equity/equality is mentioned in several places in the paper but without a concrete example for the reader. Consider adding these in as diversity, inclusion, and equity are all important considerations.
p. 7, line 304: Are there examples of co-design in the literature that could be offered specifically related to scale-up (instead of intervention development)?
p. 8, line 359: My read of the paragraph is that the crux of the issue is reach without knowing impact- is the heading appropriate or should it be ‘reach at scale with impact’?
Line 376: Revisit section 3.3 on method- what could be suggested to collect both reach and effectiveness data at scale?
p. 8, line 378: Is there some overlap with alignment of the intervention for scale-up, government priorities, and funding? Would this section fit under 3.5 (partnerships/funding)? How is it a paradox?
p. 9, lines 430+: The summary of conclusions and ‘state of affairs’ with scale-up is appreciated in the discussion. However, how do the observations from the expert group of workshop participants build on the barriers to scale-up already identified by others (and as cited in the introduction)? What can be offered as next steps or starting places to those readers interested in such work? For example, where should researchers look to draw upon frameworks/theory for scale up (issue 1)? Would RE-AIM (e.g., Glasgow et al., 2019; doi: 10.3389/fpubh.2019.00064) or another framework provide guidance on of data sources for different stakeholders (issues 2, 3)? What guidance is there on building relationships with partners (issue 4)? While part of the message is that there is no one way/approach for scale-up research, I’m left wondering where we go from here beyond recognizing the lack of current guidance.
Also consider these additional papers and relevance:
Leeman et al. (2022). Scaling up public health interventions: Engaging partners across multiple levels. Anaual Review of Public Health, 43, 155-171. https://www.annualreviews.org/doi/pdf/10.1146/annurev-publhealth-052020-113438
Weber, P.; Birkholz, L.; Kohler, S.; Helsper, N.; Dippon, L.; Ruetten, A.; Pfeifer, K.; Semrau, J. Development of a Framework for Scaling Up Community-Based Health Promotion: A Best Fit Framework Synthesis. Int. J. Environ. Res. Public Health 2022, 19, 4773. https:// doi.org/10.3390/ijerph19084773
Reviewer 3 Report
Many thanks for submitting this paper which I enjoyed reading greatly. It raises many important matters of interest to researchers, practitioners, and policymakers globally and will likely be read and cited widely. Given its importance and significance, I would suggest strongly that the authors emphasise even more explicitly the contribution of the paper to knowledge at the outset and in the conclusion. There is certainly space for more papers of this kind, and for future analyses of the tensions and paradoxes identified here. To that end, given the success of the workshop in exploring these issues, might it be worth including a little more detail on how the approach to workshop design, delivery and evaluation can support the work of others wishing to replicate the approach you took? Expressed differently, what value does this approach to coproduction have for others interested in scaling-up PA interventions, or other areas of PA work?
Opportunities for minor clarification / elaboration on the specific points raised are identified in the very minor revisions/edits listed below, but these are very minor and should not take long to address.
Abstract – worth emphasizing more explicitly the originality of the paper and its contribution to knowledge
no need for numbers in parentheses on lines 53, 55 and 61 or associated headings
57 – no need for comma
80 – funding and other resources (e.g. human, facilities, other infrastructure)?
82 – comma not semi-colon
114 – ‘for’ rather than ‘on’
148 – comma after ‘Second’
165-66 – ‘Yet in physical activity, a systems thinking approach was considered achievable’ – it is worth explaining why this is the case, and how this could be accomplished
221 – PA participation?
239 – ‘explained’ rather than ‘described’?
303-304- this is an interesting and important point. It is worth expanding a little on why this is the case
330 – delete ‘is’
382 – close up space in ‘scaling- up’
454 – close-up space in ‘research- practice’
495 – check for consistency in presentation of references in house style, especially titles of journal articles
I hope these comments assist the author in strengthening what is already an excellent paper which makes an important contribution to knowledge in the field and which should be published subject to some very minor revision. I look forward to seeing the paper in press. Thank you for taking the time to put this paper together, and congratulations on this important addition to the field.
